# Common and Distinct Functional Connectivity of the Orbitofrontal Cortex in Depression and Schizophrenia

**DOI:** 10.3390/brainsci13070997

**Published:** 2023-06-27

**Authors:** Huan Huang, Bei Rong, Cheng Chen, Qirong Wan, Zhongchun Liu, Yuan Zhou, Gaohua Wang, Huiling Wang

**Affiliations:** 1Department of Psychiatry, Renmin Hospital of Wuhan University, Wuhan 430060, China; lexie_hh@163.com (H.H.); rongbei@whu.edu.cn (B.R.); cchengwhdx@163.com (C.C.); wanqirong@whu.edu.cn (Q.W.); zcliu6@whu.edu.cn (Z.L.); wgh6402@163.com (G.W.); 2Institute of Psychology, Chinese Academy of Sciences, Beijing 100101, China; zhouyuan@psych.ac.cn; 3Hubei Institute of Neurology and Psychiatry Research, Wuhan 430060, China; 4Department of Psychiatry, Zhongxiang Hospital of Renmin Hospital of Wuhan University, Zhongxiang 431900, China; 5Hubei Provincial Key Laboratory of Developmentally Originated Disease, Wuhan 430071, China

**Keywords:** schizophrenia, depression, resting-state functional magnetic resonance imaging, intrinsic connectivity contrast, functional connectivity

## Abstract

Schizophrenia and depression are psychiatric disorders with overlapping clinical and biological features. This study aimed to identify common and distinct neuropathological mechanisms in schizophrenia and depression patients using resting-state functional magnetic resonance imaging (fMRI). The study included 28 patients with depression (DEP), 29 patients with schizophrenia (SCH), and 30 healthy control subjects (HC). Intrinsic connectivity contrast (ICC) was used to identify functional connectivity (FC) changes at the whole-brain level, and significant ICC differences were found in the bilateral orbitofrontal cortex (OFC) across all three groups. Further seed-based FC analysis indicated that compared to the DEP and HC groups, the FC between bilateral OFC and medial prefrontal cortex (MPFC), right anterior insula, and right middle frontal gyrus were significantly lower in the SCH group. Additionally, the FC between right OFC and left thalamus was decreased in both patient groups compared to the HC group. Correlation analysis showed that the FC between OFC and MPFC was positively correlated with cognitive function in the SCH group. These findings suggest that OFC connectivity plays a critical role in the pathophysiology of schizophrenia and depression and may provide new insights into the potential neural mechanisms underlying these two disorders.

## 1. Introduction

Schizophrenia and depression are highly significant global mental health problems, with substantial socioeconomic implications; however, the etiology and pathogenesis of the two disorders have not been clearly defined. Despite being distinct disease entities based on all current diagnostic criteria, there is a considerable degree of overlap in terms of genetic risk factors [1], neurobiochemical characteristics [2], immunological processes [3], and clinical magnifications [4]. These findings suggest the existence of both transdiagnostic and disease-specific neurobiological mechanism for these two disorders, which are still poorly comprehended.

Over the last two decades, the application of neuroimaging techniques has significantly contributed to our understanding of the multiple brain alterations that accompany both schizophrenia and depression. Among these techniques, resting-state functional magnetic resonance imaging (rs-fMRI) is an effective and non-invasive neuroimaging method that has emerged as a promising tool in clinical research. It allows for the assessment of spontaneous neural activity and functional organization of the brain, providing a better understanding of the underlying neural basis of individuals with mental disorders, including those with schizophrenia and depression [5,6]. Functional connectivity (FC) is a crucial aspect in the analysis of rs-fMRI, which measures temporal correlations in spontaneous low-frequency fluctuations between discrete brain regions [7]. Studies using rs-fMRI have demonstrated the hierarchical organization of the human brain into large-scale networks referred to as resting state networks (RSNs). These networks are characterized by their spatiotemporal configuration and functional roles, including the default mode network (DMN), salience network (SN), central executive network (CEN), attentional network (AN), and others [8,9]. Previous evidence has suggested that FC aberrations in several networks are relevant to both schizophrenia and depression, which are increasingly recognized as dysconnectivity disorders of brain networks [10,11,12,13]. However, few studies have directly compared FC patterns of schizophrenia and depression using rs-fMRI, and the results have been inconsistent and inconclusive. For example, one research team found that the FC between DMN and CEN was increased in schizophrenia, while it was decreased in depression [12]. Additionally, functional hypoconnectivity between the DMN and the SN was more pronounced in depression than in schizophrenia [14]. Another study demonstrated a common reduction in FC between the posterior DMN and bilateral superior parietal lobe in schizophrenia and depression, as well as diagnosis-specific connectivity reductions in the parietal operculum in schizophrenia relative to depression [13].

Therefore, the use of rs-fMRI to directly compare FC changes in patients with schizophrenia and depression can provide valuable insights into the common and unique underlying pathophysiology of these two disorders. In this study, we used a voxel-based metric called intrinsic connectivity contrast (ICC) to examine differences in resting state FC networks across the entire brain in patients with schizophrenia and depression. This approach does not rely on prior assumptions about potential differences in brain regions and allows for an unbiased search for FC abnormalities. We then analyzed the whole-brain FC patterns of the regions with altered ICC as the seeds. Our hypothesis was that patients with schizophrenia and depression would display both common and distinct FC patterns that could contribute to the neuropathology of these two diseases.

## 2. Materials and Methods

### 2.1. Participants

This study was conducted at the Department of Psychiatry, Renmin Hospital of Wuhan University, Wuhan, China, where thirty individuals diagnosed with major depressive disorder (DEP group) and thirty individuals diagnosed with schizophrenia (SCH group) were enrolled as inpatients. Additionally, thirty well-matched healthy control subjects (HC group) were included from the hospital staff and local community. Diagnosis of major depressive disorder or schizophrenia was independently determined by two experienced clinical psychiatrists based on the DSM-IV Disorder-Clinical Version (SCID-CV). Patients diagnosed with schizophrenia were in their acute phase of psychosis and were free from any current or past manic or depressive episodes, bipolar disorder, or major depression. In contrast, patients diagnosed with major depressive disorder had recurrent major depression with a current depressive episode and were free from current or past psychotic symptoms, schizoaffective disorder, schizophrenia, or bipolar disorder. Most patients were administered medication as per the established medication regimes prescribed by their psychiatrists.

Individuals who had neurologic disorders, organic mental disorders, other serious physical illnesses, dementia, brain injuries, presented substance abuse or addiction, or contraindications to MRI were not included in the study groups. Three participants were excluded from the data analyses due to artefacts, leaving a total of 30 healthy controls, 28 patients with depression and 29 patients with schizophrenia included in the analyses.

The Ethics Committee of Renmin Hospital of Wuhan University approved the study, and all participants provided their written informed consent individually after a detailed description of the study’s purpose and procedures.

### 2.2. Clinical and Cognitive Assessments

On the designated day of MRI scanning, the severity of symptoms was evaluated for patients with schizophrenia using the positive and negative syndrome scale (PANSS) [15], and for patients with depression using the 17-item Hamilton depression rating scale (HAMD-17) [16] and Hamilton anxiety rating scale (HAMA) [17]. Additionally, to determine whether patients with depression had previously experienced any symptoms related to bipolar disorder, the self-rating hypomania checklist (HCL-32) [18] was employed. For all participants, cognitive function was also assessed through the digit symbol test (DST) [19], digit span test (DSPT, both forward and backward) [20], and verbal fluency test (VFT) [21].

### 2.3. Imaging Data Acquisition

All MRI data were collected by experienced radiology professionals at the Department of Radiology, Renmin Hospital of Wuhan University, using a 3.0 T General Electric (GE) Signa HDxt MR scanner. A gradient echo planar imaging (EPI) sequence was utilized to acquire functional images (scan parameters: TR = 2000 ms, TE = 30 ms, flip angle = 90°, FOV = 220 mm × 220 mm, matrix = 64 × 64, slices = 32, slice thickness = 4 mm, gap = 0.6 mm) with each run lasting 8 min (240 volumes to be obtained). Furthermore, high-resolution T1-weighted structural images were acquired using a magnetization-prepared rapid-acquisition gradient echo sequence (scan parameters: TE = 7.8 ms, TR = 3.0 ms, flip angle = 7°, inversion time = 1100 ms, FOV = 256 mm × 256 mm, matrix = 256 × 256, slices = 188, voxel size = 1 mm × 1 mm× 1 mm).

### 2.4. Data Preprocessing

The MRI data were preprocessed and analyzed using the SPM12-based (Statistical Parametric Mapping; https://www.fil.ion.ucl.ac.uk/spm/software/ (accessed on 13 January 2020)) CONN v.19c functional connectivity toolbox (www.nitrc.org/projects/conn (accessed on 13 March 2020)) [22], which runs on MATLAB 2013b (MathWorks, Natick, MA, USA). The first five volumes of functional images were removed to account for the subjects’ adaptation to the scanning environment and steady-state magnetization. A default preprocessing pipeline for volume-based analyses was applied, including functional realignment and unwrap, functional slice-timing correction, functional direct segmentation and normalization to Montreal Neurological Institute (MNI) space, structural segmentation and normalization to MNI space, and functional smoothing using a 6 mm full-width at half-maximum (FWHM) isotropic Gaussian kernel. Outliers (for scrubbing motion and spiking artifacts) were detected using the artifact detection toolbox (ART, http://www.nitrc.org/projects/artifact_detect (accessed on 10 October 2015)). An image was defined as an outlier if the composite motion relative to the previous timepoint was greater than 2 mm or if the global mean signal intensity was exceeding 3 standard deviations from the mean image intensity, and the identified outliers were subsequently included as nuisance regressors in the first-level general linear model.

The anatomical component-based noise correction (CompCor) strategy was used for denoising, which estimates and removes motion and physiological noise without regressing out the global signal [23]. The CompCor regressors included the first five principal components (PCA) attributable to each individual’s white matter (WM) signal, the first five PCA to individual cerebrospinal fluid (CSF) signals, six subject-specific realignment parameters (three translations and three rotations) as well as their first-order temporal derivatives, the artifacts identified using ART, and the main effect of the rest scanning condition. Then, linear detrending and a temporal band-pass filter of 0.008–0.09 Hz were also applied to the data.

### 2.5. Intrinsic Connectivity Contrast Analysis

Intrinsic connectivity contrast (ICC) [24] is a measure of network centrality that is calculated on a voxel-wise basis. It characterizes the strength of the connectivity between a particular voxel and the rest of the brain. The ICC score is computed by taking the root mean square of the correlation coefficient scores between the given voxel and all of the other voxels in the brain, with higher scores indicating a stronger connectivity strength between a given voxel and every other voxel in the brain. When compared with other connectivity indexes, ICC is advantageous because it considers not only the presence of a connection, but also its strength. Additionally, it can be computed without applying a correlation threshold, meaning that it does not require any a priori information or assumptions. The ICC score was calculated for each voxel in the brain using the CONN toolbox, producing a whole-brain map for each subject. Since the biological meaning of estimating FC of the WM voxels from BOLD data is pointless, the ICC analysis was limited to grey matter (GM) voxels with an a priori GM mask (isotropic 2 mm voxels). Then, for statistical purposes, each subject’s ICC map was normalized by converting it to Z scores to fit a Gaussian distribution with zero mean and unitary variance. A higher ICC score indicates greater average strength of the correlations in a given voxel. This method has been previously used in other studies to explore FC changes without a hypothesis [25,26].

### 2.6. Seed-to-Voxel Functional Connectivity Analysis

To better understand which brain regions were involved in the observed differences in ICC, a second-level analysis was conducted. This involved selecting the brain regions that showed differences in ICC analysis between groups as seed regions, and then examining their connectivity with the rest of the brain. Pearson’s correlation coefficients were computed between the mean time series of the seed region and the time series of every other voxel in the gray matter of the whole brain. To make the data more normally distributed, a Fisher’s r-to-z transformation was applied. The resulting functional connectivity maps for each seed region were then obtained for each individual.

### 2.7. Statistical Analysis

The spatial distribution of the ICC map of each of the three groups was analyzed using one-sample *t*-tests (a voxel-wise *p* < 0.001, uncorrected, and a cluster-wise threshold of *p* < 0.05, FDR-corrected). To investigate the effect of groups and differences between groups, one-way analysis of covariance (ANCOVA) and post-hoc least significant difference (LSD) tests were performed on the ICC and seed-to-voxel FC maps controlling for age, gender, years of education, and head-motion covariable (FD value). Results were considered significant at a voxel-wise *p* < 0.001, uncorrected, and a cluster-wise threshold of *p* < 0.05, FDR-corrected.

To further examine the correlations between FC and clinical and cognitive variables, correlation analyses were performed. The altered ICC and FC scores were defined and extracted based on the results of the above ICC and seed-to-voxel FC analyses. As most variables did not meet the normality distribution assessed using Kolmogorov–Smirnov tests (*p* < 0.1), Spearman correlation analyses were used. The correlative relationship was considered to be significant at *p* < 0.05 (two-tailed) in this exploratory analysis. Subsequently, the multilevel mixed models analysis was performed to further investigate the associations while considering the grouping factor.

## 3. Results

### 3.1. Demographic, Clinical and Behavioral Data

Demographic variables across the three groups were compared with Welch’s one-way analysis of variance (ANOVA) or Mann–Whitney U test for continuous variables and Chi-square tests for categorical variables. Participants in the healthy control (HC), depression (DEP), and schizophrenia (SCH) groups were well matched in terms of age, gender, and years of education (*p* < 0.05). Cognitive function test scores were also subjected to Welch’s ANOVA analysis, exposing significant main effects of the group for all of the cognitive function tests. Post-hoc analyses indicated a considerable reduction in digit symbol test (DST), digit span test (DSPT), and verbal fluency test (VFT) scores among both patient groups when compared to the HC group, with a more marked decrease observed in the SCH group (*p* < 0.05). A detailed breakdown of the results can be observed in Table 1.

### 3.2. Intrinsic Connectivity Contrast Results

The regions exhibiting high ICC scores demonstrated a roughly symmetrical distribution and were located in the prefrontal cortex, posterior cingulate cortex/precuneus (PCC/Pcu), inferior parietal gyrus, and anterior insula in the HC group, as shown in Figure 1. This distribution is similar to the cortical hubs previously identified in ICC studies [26,27,28]. The spatial distribution of ICC in the DEP and SCH groups were generally consistent with the HC group, as shown in Figure 1.

Noteworthy group differences were discovered in the bilateral orbitofrontal cortex (OFC); both the left and right OFC, as illustrated in Figure 2 and Table 2. To further specify particular modifications between groups, parameter estimates from the right and left OFC were obtained for post-hoc LSD analyses. Outcomes showed that the left OFC connectivity was reduced in both the SCH and DEP groups relative to the HC group. Furthermore, the connectivity of the right OFC was specifically reduced in the SCH group, but not in the DEP group when compared to the HC group.

### 3.3. Seed-to-Voxel Functional Connectivity Results

We conducted a one-way ANCOVA with group as the between-subject factor. The results showed that there was a significant main effect of group in several brain regions. Specifically, the medial prefrontal cortex (MPFC) was affected when using the left orbitofrontal cortex (OFC) as a seed, while the MPFC, right anterior insula (AI), right middle frontal gyrus (MFG), and left thalamus were affected when using the right OFC as a seed. These findings are presented in Figure 3 and Table 3.

After conducting a post-hoc LSD analysis using the left OFC as a seed, we found that the SCH group had reduced FC with the MPFC compared to the DEP and HC groups. When using the right OFC as a seed, post-hoc analysis showed the SCH group demonstrated reduced FC with MPFC, rAI, and rMFG compared to the DEP and HC groups. Additionally, both the SCH and DEP group exhibited enhanced FC with the left thalamus compared to the HC group, and the FC increase between the right OFC and left thalamus was more significant in the SCH group (Figure 4).

### 3.4. Correlations between FC and Clinical Variables

Spearman correlation analyses revealed positive correlations between lOFC-MPFC FC and DST (r = 0.4094, *p* = 0.0274), as well as DSPT (forward) (r = 0.4873, *p* = 0.0073), specifically within the SCH group. The multilevel mixed models analysis demonstrated a significant fixed effect of lOFC-MPFC FC on DST (*p* = 0.0051), indicating that the strength of this connectivity is related to differences in DST performance. However, the effect of lOFC-MPFC FC on DSPT (forward) did not reach statistical significance (*p* = 0.0617) (Figure 5).

Furthermore, we found that the slopes of the relationship between lOFC-MPFC FC and DST differed significantly between the HC and DEP group (*p* < 0.05), but not significantly between the HC and SCH group or between the DEP and SCH group (*p* > 0.05). In contrast, the slopes of the relationship between lOFC-MPFC FC and DSPT (forward) showed significant differences across all three groups (*p* < 0.05).

## 4. Discussion

The current study utilized data-driven ICC analysis to examine alterations in whole-brain intrinsic functional connectivity in individuals with schizophrenia and depression and yielded several significant findings. Firstly, regions with decreased ICC scores were identified in the OFC in both the schizophrenia and depression groups. Specifically, left OFC connectivity was decreased in both groups, while right OFC connectivity was decreased only in the schizophrenia group. Secondly, functional dysconnectivity was observed between the bilateral OFC and the MPFC, as well as between right OFC and rAI and rMFG, in individuals with schizophrenia, but not in those with depression. Thirdly, both the schizophrenia and depression groups exhibited increased FC between the right OFC and left thalamus, but to varying degrees. Finally, correlation analyses indicated that the FC between the left OFC and MPFC was associated with cognitive function in the schizophrenia group. Overall, these findings provide evidence for both shared and distinct neurofunctional underpinnings of schizophrenia and depression.

The key finding of our study using the ICC analysis is the alteration observed in the bilateral OFC. Further analyses indicate a significant decrease in ICC scores in the left OFC in both schizophrenia and depression, while a significant decrease in ICC scores in the right OFC was only observed in schizophrenia but not depression. The OFC is responsible for processing and evaluating almost all types of sensory stimuli from other cortical regions, integrating them based on their associations with current needs; and it is a main site for cortico–cortico and thalamo–cortico integration. There are various functions for the OFC, including sensory integration and reward learning, cognitive flexibility, control of emotion, decision making, social behavior, and mnemonic functions [29,30,31]. The ICC analysis is a data-driven whole-brain voxel-based measure of global connectivity and qualitatively addresses different questions about brain connectivity than seed-based analysis. It evaluates the integrative capacity of a given brain region [32], reflecting each voxel’s overall connectivity strength. Our results showed the OFC exhibited concurrent reductions in integrative capacity, indicating disturbed information processing in schizophrenia and depression. Although the ICC analysis is based on FC, it cannot provide information regarding directionality; computational modeling has confirmed that nodes with a high degree (high ICC scores) tend to be the target of information flow from nodes with a lower degree [33]. Therefore, the reduced OFC ICC scores may suggest that the OFC receives less information from the other brain regions in schizophrenia and depression, leading to a decrease in the informational content and diversity, with the impairment being more remarkable in schizophrenia than depression. Our findings are consistent with previous reports in schizophrenia [34,35,36] and depression [29,37]. Moreover, using functional connectivity density (FCD) analysis combined with multivariate pattern analysis (MVPA), Chen et al. identified the OFC as the key region that could differentiate depression patients from schizophrenia patients [10]. Together, these findings provide evidence that the OFC may be relevant to the pathogenesis of schizophrenia and depression, but perhaps in different ways.

Interestingly, we found the patients with schizophrenia specifically exhibited reduced connectivity between the bilateral OFC with MPFC compared to both HC and the DEP group. This suggests that there may be disease-specific neurofunctional impairments in schizophrenia. The MPFC shares extensive anatomic connections with the OFC, and these two regions are part of a “visceromotor network” that is thought to modulate endocrine, autonomic, behavioral, and experiential aspects of emotional behavior [29,38]. In addition, the MPFC is believed to be the neural correlate of self-referential processing and also represents a core hub of the brain’s anterior DMN, which is consistently impaired in schizophrenia [39]. Both the OFC and MPFC are located in the PFC of the human brain [40], so the decoupling between OFC and MPFC was in line with previous studies and supports the hypothesis of PFC dysconnectivity in schizophrenia [41,42]. The PFC is involved in almost all cognitive functions and is essential for the organization and control of goal-directed thought and behavior [43]. Our results showed positive relationships between left OFC-MPFC FC and DST and DSPT (backward) scores specifically within the patients with schizophrenia. The DST is a measure of a person’s information processing speed while the DSST is a measure of working memory, requiring maintenance of information in memory. Our results suggest that the relationship between lOFC-MPFC FC and cognitive variables varies across different groups. Specifically, the association between FC and DST differs between healthy controls and individuals with depression, while the association between FC and DSPT (forward) differs across all three groups. This findings indicate that the cognitive deficits are differently related to the OFC-MPFC connectivity in these diseases.

Furthermore, we also observed disconnections between the right OFC and rAI, as well as rMFG, which were specific to the patients with schizophrenia. The AI and MFG are cortical hubs for the SN, first proposed by Seeley et al. [44], which is responsible for recruiting relevant brain regions for bottom-up processing of sensory information. Previous research suggests that patients with schizophrenia may have salience anomalies that reflect the connectivity deficits within the SN, and dysfunctional SN activity has been reported in schizophrenia [45]. The disrupted connectivity between the OFC and SN connectivity in schizophrenia may result in disorganized salience information processing and contribute to psychotic symptoms. Overall, our findings suggest attenuated FC between the OFC and MPFC and rAI/rMFG in patients with schizophrenia, and relatively reversed FC in patients with depression, suggesting a disrupted balance between THEOFC and DMN/SN integration in schizophrenia. These findings may serve as a disease-specific alteration for schizophrenia.

Additionally, the findings from the present study demonstrate significant overlap in increased FC between OFC and left thalamus in the SCH and DEP group. Specifically, the FC between the right OFC and left thalamus was enhanced in both groups, with a prominent increase in the SCH group where the connectivity changed from negative to positive. The thalamus is a crucial neural structure that integrates sensory information and plays a critical role in recognizing and processing emotions that involve multiple senses. Therefore, its role in mediating the complex interplay between sensory inputs and emotional responses is of paramount importance, and it has been implicated in various mental disorders [46]. Indeed, shared abnormalities in the thalamus have been reported in both schizophrenia and depression [47,48,49]. A study revealed that, compared with the healthy controls, the patients with major depression and schizophrenia showed convergent altered functional connectivity patterns related to the thalamus [47]. Our findings are consistent with these results and further suggest that the impairment of thalamic connectivity is more severe in schizophrenia than depression.

This study has some limitations that should be acknowledged. First, most patients included in the study were taking various medications for their psychotic or depressive symptoms, which may have different effects on functional connectivity as previous studies showed. This could potentially act as a major confounding factor in transdiagnostic studies. Second, it is very important to take into account the dynamics of neural stability in both clinical and non-clinical populations when interpreting the results [50]. Third, the study utilized a cross-sectional design, and all patients were at their acute episode, which limits our ability to investigate how functional connectivity may change over time. Longitudinal follow-up studies are necessary to fully understand the implications of the current findings. Additionally, the sample size in each group was relatively small, which may have led to unreliable results. Future studies with a larger number of participants are needed to increase the reliability and sensitivity of the findings. Lastly, potential confounding effects caused by head motion may exist. Previous research has identified a correlation between reduced distant functional connectivity and high head motion. However, in this study, we did not find any correlation between ICC and FC scores in the areas where between-group differences were found. We included head motion parameters as a covariate in our analysis, indicating that it is not driving the connectivity differences observed in this study.

## 5. Conclusions

To summarize, this study emphasizes the significance of OFC connectivity in the development of schizophrenia and depression. Specifically, the study found a distinct disconnection pattern between OFC and DMN/SN in schizophrenia, while both disorders showed enhanced but varying degrees of OFC–thalamus FC. These findings offer insights into the potential neural mechanisms that underlie these two psychiatric conditions.

## Figures and Tables

**Figure 1 brainsci-13-00997-f001:**
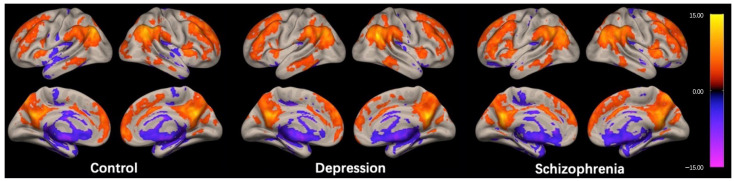
Within-group intrinsic connectivity contrast (ICC) maps for the control, depression, and schizophrenia group. The spatial patterns represent average maps of higher (red) and lower (blue) ICC scores, with a greater score representing greater average strength of the correlations in a given voxel. Statistical maps are rendered on a surface template and are thresholded voxel-wise at a *p* < 0.001, uncorrected, and a cluster-wise threshold of *p* < 0.05, FDR-corrected (one-sample *t*-test); the colorbar represents T values.

**Figure 2 brainsci-13-00997-f002:**
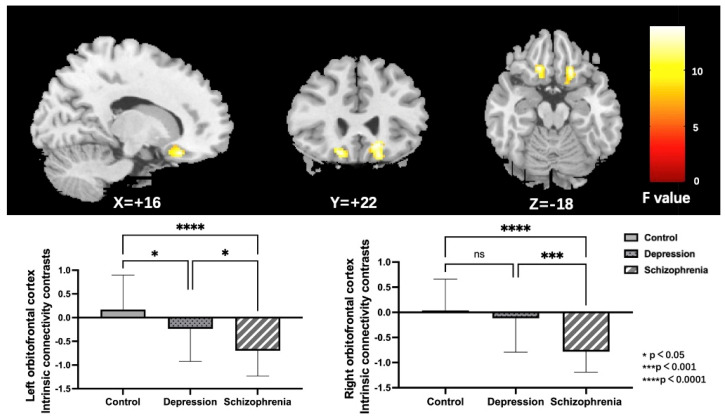
Regions (the bilateral orbitofrontal cortex) exhibiting significant group differences of ICC in the HC, DEP, and SCH groups using ANCOVA (voxel-wise *p* < 0.001, uncorrected, and a cluster-wise threshold of *p* < 0.05, FDR-corrected); the colorbar represents F values; ns: not significant.

**Figure 3 brainsci-13-00997-f003:**
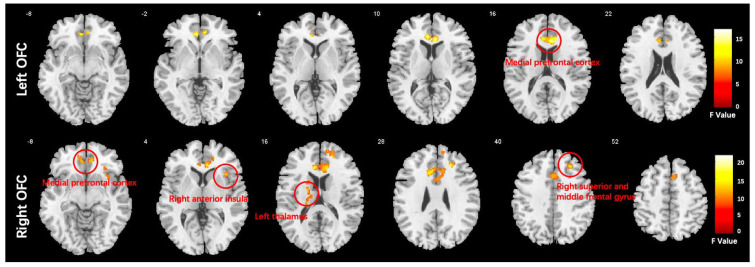
Functional connectivity differences in the orbitofrontal cortex (OFC) of the HC, DEP, and SCH groups analyzed using ANCOVA (voxel-wise *p* < 0.001, uncorrected, and a cluster-wise threshold of *p* < 0.05, FDR-corrected); the colorbar represents F values.

**Figure 4 brainsci-13-00997-f004:**
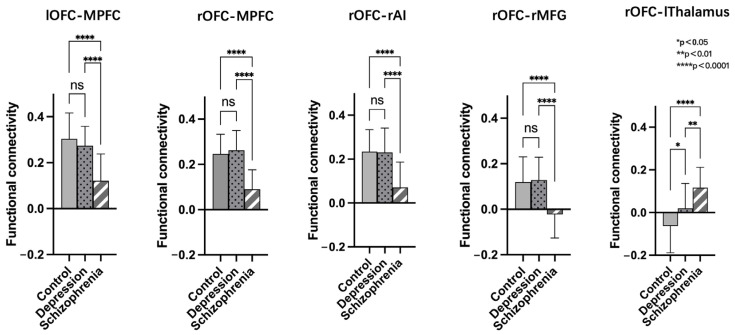
Post-hoc comparisons of functional connectivity differences in the bilateral orbitofrontal cortex (OFC) of the HC, SCH, and DEP group; ns: not significant.

**Figure 5 brainsci-13-00997-f005:**
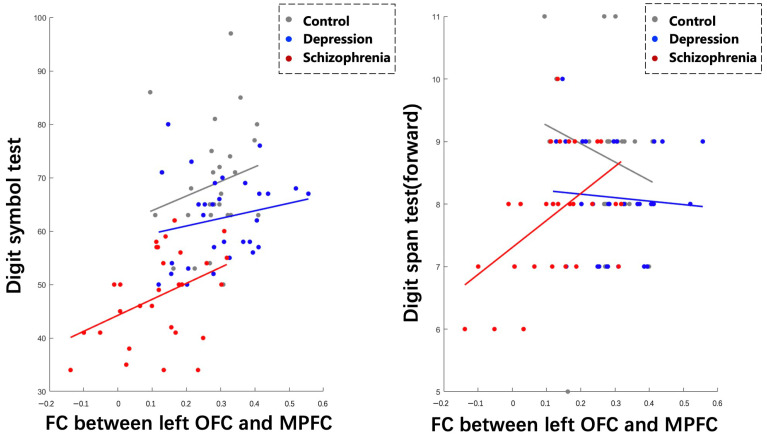
Correlations between abnormal functional connectivity and cognitive performance across three groups.

**Table 1 brainsci-13-00997-t001:** Demographic and clinical characteristic of all participants.

	Healthy Controls	Depression	Schizophrenia	*p* Value
	(HC, *n* = 30)	(DEP, *n* = 28)	(SCH, *n* = 29)	
Gender (male/female)	11/19	11/17	11/18	χ^2^ = 0.042, *p* = 0.979
Age (years)	28.37 ± 5.08	27.86 ± 6.23	27.48 ± 4.61	W = 0.246, *p* = 0.783
Education (years)	14.27 ± 2.16	14.42 ± 2.28	14.21 ± 2.74	W = 0.0.056, *p* = 0.945
Duration of illness (months)	-	25.5 (13.50, 37.75)	24.6 (12.00, 47.00)	*p* = 0.9148
Hamilton depression scale (HAMD)	-	22.32 ± 2.09	-	-
Hamilton anxiety scale (HAMA)	-	11.23 ± 4.72	-	-
Positive and negative syndrome scale (PANSS)				
Total score	-	-	83.87 ± 8.12	-
Positive scale score	-	-	21.26 ± 4.51	-
Negative scale score	-	-	20.07 ± 4.01	-
General psychopathology	-	-	41.43 ± 7.59	-
Cognitive function tests				
Digit symbol test	68.00 ± 10.8	62.61 ± 8.02	47.86 ± 8.56	W = 34.68, *p* = 0.000
Digit span test (forward)	8.77 ± 1.19	8.04 ± 0.84	7.83 ± 1.04	W = 5.617, *p* = 0.006
Digit span test (backward)	5.67 ± 1.54	5.11 ± 0.83	4.45 ± 1.12	W = 6.501, *p* = 0.003
Verbal fluency test	22.50 ± 4.90	19.64 ± 2.25	15.28 ± 3.94	W = 21.38, *p* = 0.000
Head motion (framewise displacement)	0.11 ± 0.09	0.13 ± 0.09	0.14 ± 0.10	W = 0.776, *p* = 0.456

**Table 2 brainsci-13-00997-t002:** Regions with significant group differences with intrinsic connectivity contrast in the three groups.

Regions	MNI	The z Values of FC: Mean ± SD			ANCOVA		
	Coordinates	HC	DEP	SCH	F Value	*p* Value	Cluster Size
Left orbitofrontal cortex	−20, 26, −16	0.102 ± 0.673	−0.243 ± 0.603	−0.687 ± 0.466	13.21	<0.001	177
Right orbitofrontal cortex	16, 22, −18	0.0120 ± 0.594	−0.143 ± 0.637	−0.743 ± 0.384	13.91	<0.001	327

Abbreviations: MNI = Montreal Neurological Institute; SD = standard deviation; ANCOVA = analysis of covariance; HC = healthy controls; DEP = depression; SCH = schizophrenia.

**Table 3 brainsci-13-00997-t003:** Regions with significant group differences in functional connectivity of the bilateral orbitofrontal cortex.

Regions	MNI	The z Values of FC: Mean ± SD			ANCOVA		
	Coordinates	HC	DEP	SCH	F Value	*p* Value	Cluster Size
Seed: left orbitofrontal cortex							
Medial prefrontal cortex	6, 32, 12	0.303 ± 0.113	0.273 ± 0.085	0.121 ± 0.117	15.95	<0.001	501
Seed: right orbitofrontal cortex							
Medial prefrontal cortex	8, 32, 10	0.246 ± 0.086	0.263 ± 0.087	0.090 ± 0.88	20.04	<0.001	1542
Right anterior insula	34, 14, −14	0.234 ± 0.100	0.230 ± 0.111	0.071 ± 0.115	14.92	<0.001	236
Right middle frontal gyrus	22, 34, 30	0.119 ± 0.112	0.128 ± 0.100	−0.023 ± 0.104	17.95	<0.001	181
Left thalamus	−20, −24, 20	−0.063 ± 0.126	0.0193 ± 0.117	0.117 ± 0.095	13.43	<0.001	126

Abbreviations: MNI = Montreal Neurological Institute; SD = standard deviation; ANCOVA = analysis of covariance; HC = healthy controls; DEP = depression; SCH = schizophrenia.

## Data Availability

Data can be made available upon reasonable request.

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
