# Peer review of "Common and Distinct Functional Connectivity of the Orbitofrontal Cortex in Depression and Schizophrenia"

_brainsci, 2023, doi:10.3390/brainsci13070997_

Round 1
Reviewer 1 Report
This is a classic paper on the study of connectivity networks with fMRI. This is a comparison between two groups of patients and a third control group. The work is correct, what is presented is not, however, especially novel. There are no contributions in the work that can be considered highly novel. The work is correct but of little interest in its novelty.
Furthermore, it does not adequately resolve the use of the correlations with the fMRI signal values since the p values have not been corrected and there are errors in the use of p < .05, which is contradictory to the tables. For all these reasons, in my opinion, the work does not provide relevant elements.
Reviewer 2 Report
This study uses a voxel-based metric called intrinsic connectivity contrast (ICC) to analyze resting-state functional connectivity networks in the whole brain. The objective was to compare and identify differences in FC between healthy subjects and patients with schizophrenia and depression. Significant differences were observed in the bilateral OFC across all three groups. Further analysis revealed that in the SCH group, the FC between bilateral OFC and the medial prefrontal cortex, right anterior insula, and right middle frontal gyrus was significantly lower compared to the DEP and HC groups. Additionally, the FC between the right OFC and left thalamus was reduced in both patient groups compared to the HC group. Correlation analysis indicated a positive relationship between the FC of OFC and MPFC and cognitive function in the SCH group.
The methodology employed in this study is commendable and contributes to the understanding of functional connectivity in depression and schizophrenia. The utilization of functional magnetic resonance imaging to investigate the common and distinct connectivity patterns of the OFC is a robust approach. The use of a voxel-based metric such as ICC adds depth to the analysis and allows for a comprehensive examination of FC networks throughout the entire brain.
The inclusion of both patient groups, namely those with depression and schizophrenia, provides valuable insights into the unique functional connectivity characteristics associated with each disorder. Additionally, the comparison between these patient groups and a healthy control group enhances the validity of the findings and supports the identification of specific alterations in FC.
The study's focus on the OFC, a region implicated in emotion processing and cognitive control, is appropriate and relevant to understanding the pathophysiology of depression and schizophrenia. The findings obtained through seed-based FC analysis contribute to our knowledge by identifying significant connectivity differences between the OFC and other brain regions, such as the MPFC, anterior insula, and middle frontal gyrus, in both patient groups.
Overall, the paper is written in a correct manner and I think it could be of interest to Brain Sciences readers. Here are some suggestions that the authors might consider including:
1- In section 2.2 they use various measuring instruments. It would be appropriate to add references in case other researchers want to use the same instruments and compare their results.
2- On p4 (l 153 and 155) it says: "Additionally, it can be computed without applying a correlation threshold, meaning that it does not require any a priori information or assumptions". Since I am not familiar with the ICC, I would like the authors to explain specifically why thresholds do not need to be taken into account.
3- In the analysis section I have some doubts. For example (l 174), to compare the ICC map of the three groups they use a t-test, which is a parametric test, suitable when the assumptions of normality and homogeneity of variances are met. However, this section does not mention whether the assumptions have been studied, and I suspect that if they had been, they would not be fulfilled, so nonparametric tests would be more appropriate. However, in the paragraph from line 181 to line 187, it is mentioned that Spearman will be used, which is a nonparametric technique, since the assumption of normality is not met. The same as for the comparison of demographic, clinical and behavioral data, where F and t tests are used, and I think it would be more appropriate, if as I suspect, the assumptions are not met, to use nonparametric tests. In short, I believe that the authors could take a more coherent approach to the analyses and use non-parametric techniques.
4- It is not clear to me, why they sometimes perform a one-tailed test (e.g., l 173) and sometimes opt for two-tailed tests (e.g., l 185 or 186).
Reviewer 3 Report
The study " Common and Distinct functional connectivity of the orbito-2 frontal cortex in depression and schizophrenia" by Huang and colleagues reported interesting results concerning the differential neural signatures of individuals with depression and schizophrenia, compared with healthy controls.
The study is methodologically well-done, although I would have performed a network analysis in parallel with the ICC done by the authors to corroborate their interpretations at the network level. I am generally positive in endorsing this publication, but the authors should revise it in some points.
- The authors observed decreased ICC in the OFC, which is interpreted as the OFC in schizophrenia and depression being less integrated with the rest of the brain. Such segregation, standing at the authors, is specifically relevant between OFC and MPFC in schizophrenics since this relates with cognitive deficits.
Please report these associations (as in Figure 5) also for healthy subjects and individuals with depression. Use specific comprehensive analysis (e.g., multilevel mixed models) to compare the associations between groups. Are the slopes particularly different across groups?
Update the Discussion session according to the findings.
- Lines 146-151. How were negative correlation values handled in the calculation of the ICC?
- Please specify how the minimum cluster size and cluster thresholds were calculated? Was the structure of noise autocorrelations considered in this process?
- In the Discussion section, the authors describe in detail their findings related to schizophrenics. I understand that the study’s focus in on schizophrenic subjects, but many readers would be interested in reading considerations about the inter-individual variability also in healthy subjects.
It is known that schizophrenics exhibit less stable neural organizations (Gifford et al., 2020), but decreased stability can generally predispose healthy individuals toward psychotic conditions or self-disturbances (McGrath et al., 2015). In fact, it has been recently found that DMN instability correlates with high psychosis-like experiences in healthy individuals (Di Plinio et al., 2022). How do the authors interpret their findings also considering these dynamics in healthy subjects?
REFERENCES
Di Plinio, S, Ebisch SJH (2022). Probabilistically Weighted Multilayer Networks disclose the link between default mode network instability and psychosis-like experiences in healthy adults. NeuroImage, 119219.
Gifford, G., Crossley, N., Kempton, M.J., Morgan, S., Dazzan, P., Young, J., McGuire, P., 2020. Resting state fMRI based multilayer network configuration in patients with schizophrenia. NeuroImage: Clinical 25, 102169.
McGrath, J.J., Saha, S., Al-Hamzawi, A., et al., 2015. Psychotic experiences in the general population: a cross-national analysis based on 31,261 respondents from 18 countries. JAMA Psychiatry 72 (7), 697–705.
Round 2
Reviewer 1 Report
This new version resolves most of my previous comments.